

# Zonal-mean data set of global atmospheric reanalyses on pressure levels

Patrick Martineau[1], Jonathon S. Wright[2], Nuanliang Zhu[2,3], and Masatomo Fujiwara[4]

[1]Research Center for Advanced Science and Technology, The University of Tokyo, Tokyo, Japan
[2]Ministry of Education Key Laboratory of Earth System Modeling, Department of Earth System Science, Tsinghua University, Beijing 100084, China
[3]Department of Engineering Physics, Tsinghua University, Beijing 100084, China
[4]Faculty of Environmental Earth Science, Hokkaido University, Sapporo, 060-0810, Japan

*Correspondence to:* Patrick Martineau (pmartineau@atmos.rcast.u-tokyo.ac.jp)

**Abstract.** This data set, which is prepared for the SPARC-Reanalysis Intercomparison Project (S-RIP), provides several zonal-mean diagnostics computed from reanalysis data on pressure levels. Diagnostics are currently provided for a variety of reanalyses, including ERA-40, ERA-Interim, ERA-20C, NCEP-NCAR, NCEP-DOE, CFSR, 20CR v2 and v2c, JRA-25, JRA-55, JRA-55C, JRA-55AMIP, MERRA, and MERRA-2. The data set will be expanded to include additional reanaly-

ses as they become available. Basic dynamical variables (such as temperature, geopotential height and three-dimensional winds) are provided in addition to a complete set of terms from the Eulerian-mean and transformed Eulerian-mean momentum equations. Total diabatic heating and its long-wave and short-wave components are included as availability permits, along with heating rates diagnosed from the basic dynamical variables using the zonal-mean thermodynamic equation. Two versions of the data set are provided, one that uses horizontal and vertical grids provided by the various reanalysis centers, and

another that uses a common grid to facilitate comparison among data sets. For the common grid, all diagnostics are interpolated horizontally onto a regular 2.5°×2.5° grid for a subset of pressure levels that are common amongst all included reanalyses. The dynamical (Martineau, 2017, http://dx.doi.org/10.5285/b241a7f536a244749662360bd7839312) and diabatic (Wright, 2017, http://dx.doi.org/10.5285/70146c789eda4296a3c3ab6706931d56) variables are archived and maintained by the Centre for Environmental Data Analysis (CEDA).

**1  Introduction**

Reanalysis products are commonly used to study weather and climate variability and to validate climate models. By combining numerical forecast models and various observations through data assimilation procedures, reanalyses aim to produce a best estimate of the state of the atmosphere. However, differences among the model and assimilation components of reanalysis systems, as well as differences in the assimilated observations, result in different representations of the historical state and

behavior of the atmosphere. These discrepancies contribute to uncertainties in our understanding of the atmosphere and its variability.



The Stratosphere–troposphere Processes And their Role in Climate (SPARC) Reanalysis Intercomparison Project (S-RIP; Fujiwara et al., 2017, see also http://s-rip.ees.hokudai.ac.jp/) undertakes to compare reanalysis data sets, understand the causes of the differences, and provide guidance on the appropriate usage of reanalyses (all abbreviations are collected in Appendix A). To facilitate this comparison, we have prepared a data set containing zonal-mean variables on pressure levels using a consistent

set of numerical methods and a unified file format.

The data set comprises two major components. The first component provides variables on an 'original' latitude–pressure grid defined by the corresponding reanalysis center (Original Grid; OG). Note that this grid is typically not defined by the model resolution; nor is it necessarily unique, as some reanalysis products are distributed on a range of grids (Table 1). The second component is a data set for which basic variables have been interpolated onto a common $2.5° \times 2.5°$ latitude–longitude

grid (Common Grid, CG). The pressure coordinate for the CG data files is reduced to contain only those levels common to all of the reanalysis data sets, with extension up to 1 hPa when possible. Both data sets are provided on 6-h time intervals. While the OG zonal-mean diagnostics are affected by the horizontal grid on which variables are provided, the CG diagnostics have no such dependence and can be directly compared without further interpolation.

The characteristics of this zonal-mean data set on pressure levels are described in this paper. The reanalysis data sets included

in the comparison are listed and briefly described in Section 2. The diagnostics provided in the data set are introduced in Section 3, with grid dependence discussed in Section 4. The availability of the S-RIP zonal-mean data set and its appropriate usage are outlined in Section 5.

## 2   Data

The zonal-mean data set on pressure levels includes most major reanalysis products (Table 1), with a total of fourteen re-

analyses represented. Three of these reanalyses have been produced by the European Centre for Medium-Range Weather Forecasts (ECMWF): the ECMWF 40-year Reanalysis (ERA-40), the ECMWF Interim Reanalysis (ERA-Interim), and the ECMWF Twentieth Century Reanalysis (ERA-20C). Five of the reanalyses have been produced by the National Centers for Environmental Prediction (NCEP) and cooperating agencies: the NCEP–National Center for Atmospheric Research Reanalysis 1 (NCEP–NCAR), the NCEP–Department of Energy Reanalysis 2 (NCEP–DOE), the Climate Forecast System Reanalysis

(CFSR), and the National Oceanic and Atmospheric Administration (NOAA) and Cooperative Institute for Research in Environmental Sciences (CIRES) 20th Century Reanalysis (20CR) version 2 (v2) and version 2c (v2c). Note that CFSR products after January 2011 have been produced with the slightly different data assimilation system CFSv2 (Fujiwara et al., 2017, their Section 2.4). 20CR v2c uses the same model as 20CR v2, but with new sea ice boundary conditions, amongst other changes (Gil Compo, private communication, 2017). Four of the reanalyses have been produced by the Japan Meteorological Agency (JMA):

the Japanese 25-year Reanalysis (JRA-25), the Japanese 55-year Reanalysis (JRA-55), and two variants of JRA-55 (JRA-55C and JRA-55AMIP). JRA-55 and its variants all use the same model and boundary conditions; however, while JRA-55 assimilates both conventional and satellite observations, JRA-55C assimilates only conventional observations and JRA-55AMIP does not assimilate any observations. The final two reanalyses included in the data set have been produced by the National Aero-





nautics and Space Administration (NASA): the Modern Era Retrospective-analysis for Research and Applications (MERRA) and its successor MERRA-2.

**Table 1.** List of reanalyses represented in the S-RIP zonal-mean data set.

| Name | Label | Period provided | Reference | Highest level (hPa) | Grid resolution (°) [a] |
|------|-------|-----------------|-----------|---------------------|------------------------|
| ERA-40 | E40 | 1958-2002 | Uppala et al. (2005) | 1 | 2.5 |
| ERA-Interim | E-I | 1979-2016 | Dee et al. (2011) | 1 | 1.5 |
| ERA-20C | E20 | 1958-2010 | Poli et al. (2016) | 1 | 1.1215 |
| NCEP-NCAR | N-N | 1958-2016 | Kalnay et al. (1996) | 10 | 2.5 |
| NCEP-DOE | N-D | 1979-2016 | Kanamitsu et al. (2002) | 10 | 2.5 |
| CFSR[b] | CFS | 1979-2016 | Saha et al. (2010, 2014) | 1 | 2.5 |
| 20CR (v2) | 20CR2 | 1958-2012 | Compo et al. (2011) | 10 | 2 |
| 20CR (v2c) | 20CR2c | 1958-2014 | Compo et al. (2011) | 10 | 2 |
| JRA-25 | J25 | 1979-2013 | Onogi et al. (2007) | 1 | 2.5 |
| JRA-55 | J55 | 1958-2016 | Kobayashi et al. (2015) | 1 | 1.25 |
| JRA-55C | J55C | 1979-2012 | Kobayashi et al. (2014) | 1 | 1.25 |
| JRA-55AMIP | J55A | 1958-2012 | Kobayashi et al. (2014) | 1 | 1.25 |
| MERRA[c] | ME | 1979-2015 | Rienecker et al. (2011) | 0.1 | 1.25 |
| MERRA-2[c] | ME2 | 1980-2016 | Gelaro et al. (2017) | 0.1 | 1.25 |

[a] Original grid resolution when downloaded from the source; some reanalyses provide data on multiple grids.

[b] Transition from version 1 (CFSR) to version 2 (CFSv2) on 1 January 2011

[c] For MERRA and MERRA-2, only ASM products are used (see also discussion by Fujiwara et al., 2017).

Horizontal grid sizes in degrees for the reanalysis products used to produce the OG zonal-mean data set are listed in Table 1. All data are on regular latitude–longitude grids. Model-generated diabatic heating products from CFSR and MERRA-2 are remapped directly from the default grids ($1°$ and $0.5 \times 0.625°$, respectively) onto the OG and CG grids listed for these reanalyses in Table 1 using bilinear interpolation.

The highest pressure level provided is also listed in Table 1. The pressure levels included in each data set are shown in Table 2. As discussed below, the grid spacing does not have a large impact on the diagnostics provided in this data set. Inter-reanalysis differences in zonal-mean diagnostics are dominated by reanalysis-specific factors rather than numerical resolution. Differences amongst reanalysis products emerge from differences in the underlying models, data assimilation techniques, and assimilated observations. A detailed accounting of these differences is beyond the scope of this article. Fujiwara et al. (2017) have recently reviewed many of the technical differences amongst these reanalyses. A focused intercomparison of monthly-mean temperatures and winds conducted by Long et al. (2017) further revealed a good level of agreement among reanalyses,



particularly the most recent products, but with a time dependence that emerges from changes in the assimilated observational data.

## 3 Diagnostics

With the exception of the model-generated diabatic heating rates (see section 3.6.1), all diagnostics provided in this data set are
based on three-dimensional wind $(u, v, \omega)$, temperature $(T)$, and geopotential height $(Z)$ fields provided on levels of constant pressure $(p)$. All diagnostics are evaluated as a function of time $t$ at 6-h intervals and provided along a regularly-spaced latitude coordinate $(\phi)$. Data access information for core variables is provided in Appendix A, Table A1.

Potential temperature is calculated on pressure levels as follows:

$$\theta = T \left( \frac{p_0}{p} \right)^{R_d/c_p} \tag{1}$$

where $p_0$ is a reference pressure (1000 hPa), $R_d$ is the gas constant for dry air (287 J K$^{-1}$ kg$^{-1}$), and $c_p$ is the specific heat at constant pressure (1004 J K$^{-1}$ kg$^{-1}$). The ratio $R_d/c_p$ is rounded to 0.286. Throughout this paper, the zonal mean of a quantity $x$ is denoted as $\overline{x}$, with anomalies from the zonal mean defined as $x' = x - \overline{x}$.

Differences in the preparation of the OG and CG data sets are illustrated in Fig. 1. All calculations for the OG data set are performed on the original grid associated with that reanalysis (Table 1). For the CG data set, all variables are first interpolated
to the common 2.5°×2.5° grid using bilinear interpolation in latitude and longitude. Only common pressure levels listed in Table 2 are kept. Diagnostics are then computed on the common grid before the zonal mean is taken.

### 3.1 Numerical Methods

A three point stencil is used to evaluate all derivatives. In the case of meridional derivatives the three point stencil is expressed as

$$\frac{\partial f(\phi)}{\partial \phi} \approx \frac{f(\phi + \Delta\phi) - f(\phi - \Delta\phi)}{2\Delta\phi}. \tag{2}$$

where $\phi$ is latitude in radians. This scheme, which has an accuracy of the order of $(\Delta\phi)^2$, is chosen for its ability to evaluate derivatives close to the boundaries. In the case of vertical derivatives the three point stencil is expressed as:

$$\frac{\partial f(p)}{\partial p} \approx \frac{f(p + \Delta p) - f(p - \Delta p)}{2\Delta p}. \tag{3}$$

Since pressure levels are not evenly spaced, the centered difference in the vertical is first computed for half-levels (in the pres-
sure domain) and then linearly interpolated (still in the pressure domain) back to the original pressure levels. No extrapolation is performed; vertical derivatives at the lowermost and uppermost pressure levels are not provided.



**Table 2.** Vertical levels of the CG and OG data sets. Pressure levels provided in the OG data set are indicated with x and pressure levels provided in the CG data set are highlighted in gray.

| Level (hPa) | E40 | E-I | E20 | N-N | N-D | CFS | J25 | J55 | J55C | J55A | ME | ME2 | 20CR2 | 20CR2c |
|---|---|---|---|---|---|---|---|---|---|---|---|---|---|---|
| 0.1 | | | | | | | | | | | x | x | | |
| 0.3 | | | | | | | | | | | x | x | | |
| 0.4 | | | | | | | | | | | x | x | | |
| 0.5 | | | | | | | | | | | x | x | | |
| 0.7 | | | | | | | | | | | x | x | | |
| 1 | x | x | x | | | x | x | x | x | x | x | x | | |
| 2 | x | x | x | | | x | x | x | x | x | x | x | | |
| 3 | x | x | x | | | x | x | x | x | x | x | x | | |
| 4 | | | | | | | | | | | x | x | | |
| 5 | x | x | x | | | x | x | x | x | x | x | x | | |
| 7 | x | x | x | | | x | x | x | x | x | x | x | | |
| 10 | x | x | x | x | x | x | x | x | x | x | x | x | x | x |
| 20 | x | x | x | x | x | x | x | x | x | x | x | x | x | x |
| 30 | x | x | x | x | x | x | x | x | x | x | x | x | x | x |
| 40 | | | | | | | | | | | x | x | | |
| 50 | x | x | x | x | x | x | x | x | x | x | x | x | x | x |
| 70 | x | x | x | x | x | x | x | x | x | x | x | x | x | x |
| 100 | x | x | x | x | x | x | x | x | x | x | x | x | x | x |
| 125 | | x | x | | | x | | x | x | x | | | | |
| 150 | x | x | x | x | x | x | x | x | x | x | x | x | x | x |
| 175 | | x | x | | | x | | x | x | x | | | | |
| 200 | x | x | x | x | x | x | x | x | x | x | x | x | x | x |
| 225 | | x | x | | | x | | x | x | x | | | | |
| 250 | x | x | x | x | x | x | x | x | x | x | x | x | x | x |
| 300 | x | x | x | x | x | x | x | x | x | x | x | x | x | x |
| 350 | | x | x | | | x | | x | x | x | x | x | x | x |
| 400 | x | x | x | x | x | x | x | x | x | x | x | x | x | x |
| 450 | | x | x | | | x | | x | x | x | x | x | x | x |
| 500 | x | x | x | x | x | x | x | x | x | x | x | x | x | x |
| 550 | | x | x | | | x | | x | x | x | x | x | x | x |
| 600 | x | x | x | x | x | x | x | x | x | x | x | x | x | x |
| 650 | | x | x | | | x | | x | x | x | x | x | x | x |
| 700 | x | x | x | x | x | x | x | x | x | x | x | x | x | x |
| 725 | | | | | | | | | | | x | x | | |
| 750 | | x | x | | | x | | x | x | x | x | x | x | x |
| 775 | x | x | x | | | x | | x | x | x | x | x | | |
| 800 | | x | x | | | x | | x | x | x | x | x | x | x |
| 825 | | x | x | | | x | | x | x | x | x | x | | |
| 850 | x | x | x | x | x | x | x | x | x | x | x | x | x | x |
| 875 | | x | x | | | x | | x | x | x | x | x | | |
| 900 | | x | x | | | x | | x | x | x | x | x | x | x |
| 925 | x | x | x | x | x | x | x | x | x | x | x | x | | |
| 950 | | x | x | | | x | | x | x | x | x | x | x | x |
| 975 | | x | x | | | x | | x | x | x | x | x | | |
| 1000 | x | x | x | x | x | x | x | x | x | x | x | x | x | x |



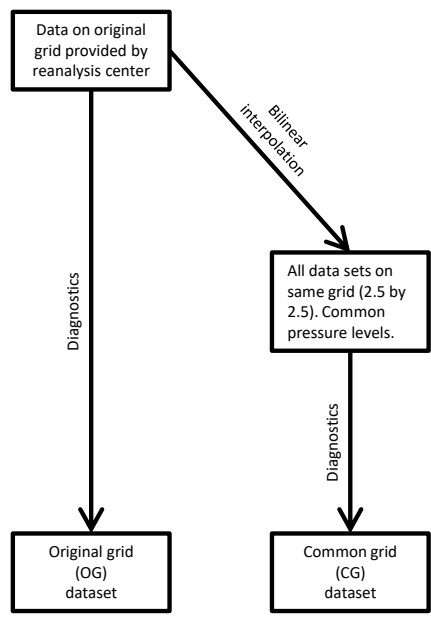

**Figure 1.** Flowchart illustrating differences in the calculation of diagnostics in the Original Grid (OG) and Common Grid (CG) data sets.

## 3.2 Core zonal-mean variables

The core of the data set consists of simple zonal-mean diagnostics. Zonal-mean variables such as zonal wind, meridional wind, temperature and geopotential height are provided (Table 3). These basic quantities are then used to produce the advanced diagnostics.

**Table 3.** Core zonal-mean variables.

| Variable | Expression | Units |
|---|---|---|
| Zonal wind | $\overline{u}$ | $\mathrm{m\,s^{-1}}$ |
| Meridional wind | $\overline{v}$ | $\mathrm{m\,s^{-1}}$ |
| Vertical wind | $\overline{\omega}$ | $\mathrm{Pa\,s^{-1}}$ |
| Temperature | $\overline{T}$ | K |
| Geopotential height | $\overline{Z}$ | m |



## 3.3 Covariance terms

Several covariance terms are provided (Table 4). The covariance between zonal wind and meridional wind (momentum flux) and the covariance of meridional wind and temperature (heat flux) are often used to assess the propagation of eddies in the zonal-mean framework. These covariance terms also enter the computation of the transformed Eulerian mean and EP flux

diagnostics (described below). The variances of zonal wind ($\overline{u'^2}$) and meridional wind ($\overline{v'^2}$) can be used to compute eddy kinetic energy as $EKE = 1/2 \left( \overline{u'^2 + v'^2} \right)$.

**Table 4.** Covariance terms.

| Variable | Expression | Units |
|---|---|---|
| Zonal wind | $\overline{u'^2}$ | $\mathrm{m^2\,s^{-2}}$ |
| Meridional wind | $\overline{v'^2}$ | $\mathrm{m^2\,s^{-2}}$ |
| Temperature | $\overline{T'^2}$ | $\mathrm{K^2}$ |
| Zonal wind and meridional wind | $\overline{u'v'}$ | $\mathrm{m^2\,s^{-2}}$ |
| Zonal wind and vertical wind | $\overline{u'\omega'}$ | $\mathrm{Pa\,m\,s^{-2}}$ |
| Vertical wind and temperature | $\overline{\omega'T'}$ | $\mathrm{Pa\,K\,s^{-1}}$ |
| Meridional wind and temperature | $\overline{v'T'}$ | $\mathrm{K\,m\,s^{-1}}$ |

## 3.4 Eulerian mean momentum diagnostics

The zonal-mean tendency of zonal wind $\frac{\partial \overline{u}}{\partial t}$ is expressed using the primitive momentum equation on pressure levels:

$$\frac{\partial \overline{u}}{\partial t} = f\overline{v} - \overline{v}\frac{1}{a\cos\phi}\frac{\partial(\overline{u}\cos\phi)}{\partial\phi} - \overline{\omega}\frac{\partial\overline{u}}{\partial p} - \frac{1}{a\cos^2\phi}\frac{\partial(\overline{u'v'}\cos^2\phi)}{\partial\phi} - \frac{\partial(\overline{\omega'u'})}{\partial p} + \overline{\epsilon}, \tag{4}$$

where $f$ is the Coriolis frequency ($f = 2\Omega\sin\phi$), $\Omega$ is the rotation rate of the Earth ($7.2921 \times 10^{-5}\,\mathrm{rad\,s^{-1}}$) and $a$ is the mean radius of the Earth ($6\,371\,000\,\mathrm{m}$) (Andrews et al. 1987; see Salby 1996 for the transformation from log-pressure coordinates to isobaric coordinates). The first five terms on the right-hand side of Equation 4 are provided as Eulerian mean momentum diagnostics (Table 5). The residual term $\overline{\epsilon}$ includes the effects of parameterized processes, diffusion, and errors in the numerical methods. This term may be evaluated by subtracting the sum of the five terms listed in Table 5 from the zonal-mean tendency

of zonal wind.





**Table 5.** Eulerian-mean momentum diagnostics.

| Variable | Expression | Units |
|---|---|---|
| Coriolis torque | $f\overline{v}$ | $\mathrm{m\,s^{-2}}$ |
| Meridional advection of momentum | $-\overline{v}\frac{1}{a\cos\phi}\frac{\partial(\overline{u}\cos\phi)}{\partial\phi}$ | $\mathrm{m\,s^{-2}}$ |
| Vertical advection of momentum | $-\overline{\omega}\frac{\partial\overline{u}}{\partial p}$ | $\mathrm{m\,s^{-2}}$ |
| Meridional momentum flux convergence | $-\frac{1}{a\cos^2\phi}\frac{\partial(\overline{u'v'}\cos^2\phi)}{\partial\phi}$ | $\mathrm{m\,s^{-2}}$ |
| Vertical momentum flux convergence | $-\frac{\partial(\overline{\omega'u'})}{\partial p}$ | $\mathrm{m\,s^{-2}}$ |

**Table 6.** TEM momentum diagnostics.

| Variable | Expression | Units |
|---|---|---|
| Coriolis torque | $f\overline{v}^*$ | $\mathrm{m\,s^{-2}}$ |
| Meridional advection of momentum | $-\overline{v}^*\frac{1}{a\cos\phi}\frac{\partial(\overline{u}\cos\phi)}{\partial\phi}$ | $\mathrm{m\,s^{-2}}$ |
| Vertical advection of momentum | $-\overline{\omega}^*\frac{\partial\overline{u}}{\partial p}$ | $\mathrm{m\,s^{-2}}$ |
| Meridional residual circulation | $\overline{v}^*$ | $\mathrm{m\,s^{-1}}$ |
| Vertical residual circulation | $\overline{\omega}^*$ | $\mathrm{Pa\,s^{-1}}$ |
| EP-flux (meridional component) | $F_\phi$ | $\mathrm{m^3\,s^{-2}}$ |
| EP-flux (vertical component) | $F_p$ | $\mathrm{Pa\,m^2\,s^{-2}}$ |
| EP-flux (meridional component, QG) | $F_\phi^{QG}$ | $\mathrm{m^3\,s^{-2}}$ |
| EP-flux (vertical component, QG) | $F_p^{QG}$ | $\mathrm{Pa\,m^2\,s^{-2}}$ |
| EP-flux divergence (meridional component) | $\frac{1}{a\cos\phi}\nabla\cdot F_\phi$ | $\mathrm{m\,s^{-2}}$ |
| EP-flux divergence (vertical component) | $\frac{1}{a\cos\phi}\nabla\cdot F_p$ | $\mathrm{m\,s^{-2}}$ |
| EP-flux divergence (vertical component, QG) | $\frac{1}{a\cos\phi}\nabla\cdot F_\phi^{QG}$ | $\mathrm{m\,s^{-2}}$ |
| EP-flux divergence (meridional component, QG) | $\frac{1}{a\cos\phi}\nabla\cdot F_p^{QG}$ | $\mathrm{m\,s^{-2}}$ |



### 3.5 Transformed Eulerian Mean (TEM) momentum diagnostics

#### 3.5.1 Primitive-equation version

Transformed Eulerian Mean (TEM) momentum diagnostics (Table 6) based on the primitive momentum equation are provided on pressure levels (e.g. Andrews et al., 1987; Salby, 1996). The residual circulation is first defined as follows:

$$\overline{v}^* = \overline{v} - \frac{\partial}{\partial p}\left[\frac{\overline{v'\theta'}}{\partial\overline{\theta}/\partial p}\right] \qquad \overline{\omega}^* = \overline{\omega} + \frac{1}{a\cos\phi}\frac{\partial}{\partial\phi}\left[\frac{\overline{v'\theta'}\cos\phi}{\partial\overline{\theta}/\partial p}\right]. \tag{5}$$

Substituting Eq. 5 into Eq. 4, we obtain the TEM equation:

$$\frac{\partial\overline{u}}{\partial t} = f\overline{v}^* - \overline{v}^*\frac{1}{a\cos\phi}\frac{\partial(\overline{u}\cos\phi)}{\partial\phi} - \overline{\omega}^*\frac{\partial\overline{u}}{\partial p} + \frac{1}{a\cos\phi}\boldsymbol{\nabla}\cdot\boldsymbol{F} + \overline{\epsilon}. \tag{6}$$

Here, the Eliassen–Palm (EP) flux is a two dimensional vector defined as:

$$\{F_\phi, F_p\} = a\cos\phi\left\{\frac{\overline{v'\theta'}}{\partial\overline{\theta}/\partial p}\frac{\partial\overline{u}}{\partial p} - \overline{u'v'}, -\frac{\overline{v'\theta'}}{\partial\overline{\theta}/\partial p}\frac{1}{a\cos\phi}\frac{\partial}{\partial\phi}(\overline{u}\cos\phi) + \frac{\overline{v'\theta'}}{\partial\overline{\theta}/\partial p}f - \overline{\omega'u'}\right\}, \tag{7}$$

with the divergence operator in spherical coordinates defined as

$$\nabla\cdot\boldsymbol{F} = \frac{1}{a\cos\phi}\frac{\partial(F_\phi\cos\phi)}{\partial\phi} + \frac{\partial(F_p)}{\partial p}. \tag{8}$$

The residual ($\overline{\epsilon}$) is mathematically equivalent to $\overline{\epsilon}$ as defined in the Eulerian-mean framework (Eq. 4).

#### 3.5.2 Quasi-geostrophic (QG) approximation

The quasi-geostrophic (QG) version of the TEM equation (Edmon et al., 1980) is expressed as:

$$\frac{\partial\overline{u}}{\partial t} = f\overline{v}^* + \frac{1}{a\cos\phi}\boldsymbol{\nabla}\cdot\boldsymbol{F^{QG}} + \overline{\epsilon_{QG}}, \tag{9}$$

where the QG EP flux takes the form:

$$\{F_\phi^{QG}, F_p^{QG}\} = a\cos\phi\left\{-\overline{u'v'}, \frac{\overline{v'\theta'}}{\partial\overline{\theta}/\partial p}f\right\}. \tag{10}$$

Although we use the QG form of the momentum equation, the resulting diagnostics are not strictly QG since the total wind is used rather than the geostrophic wind. The vertical and meridional components of the QG EP flux and EP-flux divergence are provided in addition to the primitive-equation terms (Table 6).

### 3.6 Diabatic heating rates

#### 3.6.1 Model-generated heating rates

Zonal-mean diabatic heating rates generated by a subset of the reanalysis forecast models are provided at 6-h resolution in units of K day$^{-1}$. Unlike the other diagnostics, these terms are not derived using the basic variables listed in Table 3. Instead,



these zonal-mean diabatic heating rates are computed from the physical temperature tendency diagnostics produced during the reanalysis model forecast step. This approach allows for separate analyses of total, radiative, and non-radiative heating. Not all reanalyses store these forecast products or make them publicly available. Heating rates in the S-RIP zonal-mean data set are provided for only eight of the fourteen reanalyses: ERA-40, ERA-Interim, NCEP–NCAR, CFSR, JRA-25, JRA-55,

MERRA, and MERRA-2. Heating rate forecasts were not archived for CFSv2, and are therefore only available for CFSR through December 2010. Data access information for all heating rate products is provided in Appendix A, Table A2.

**Table 7.** Model-generated diabatic heating diagnostics.

| Variable | Expression | Units |
|---|---|---|
| Total diabatic heating due to parametrized physics | $\overline{\frac{\theta}{c_p T} \dot{Q}}$ | $\mathrm{K\,day}^{-1}$ |
| Diabatic heating due to long-wave radiation | $\overline{\frac{\theta}{c_p T} \dot{Q}_{\mathrm{LW}}}$ | $\mathrm{K\,day}^{-1}$ |
| Diabatic heating due to short-wave radiation | $\overline{\frac{\theta}{c_p T} \dot{Q}_{\mathrm{SW}}}$ | $\mathrm{K\,day}^{-1}$ |

Diabatic heating is a fundamental component of the temperature budget, as expressed by the thermodynamic equation in pressure coordinates:

$$\frac{\partial \theta}{\partial t} + \mathbf{v} \cdot \nabla \theta + \omega \frac{\partial \theta}{\partial p} = \frac{\theta}{c_p T} \dot{Q}, \tag{11}$$

The terms on the left-hand side of Eq. 11 constitute the material derivative of potential temperature ($\frac{D\theta}{Dt}$). The term on the right-hand side represents diabatic heating due to physical processes, such as latent heating, radiative transfer, and vertical diffusion. Three diabatic terms are included with the OG and CG zonal-mean data sets (Table 7): total diabatic heating due to all parametrized physics, diabatic heating due to long-wave radiative transfer, and diabatic heating due to short-wave radiative transfer. These terms, which are provided by the reanalyses as temperature tendencies, are converted here to potential tempera-

ture tendencies. As the variables are provided on pressure levels, they can be easily converted back to zonal-mean temperature tendencies if desired. Note that the diabatic heating terms are based on average temperature tendencies over each 6-h window. Accordingly, these data are centered at 03, 09, 15, and 21Z rather than the standard synoptic times 00, 06, 12, and 18Z. Other diagnostics provided in the OG and CG zonal-mean data sets are based on instantaneous fields at the standard synoptic times. The diabatic heating rates thus lag these other diagnostics by 3 h.

**3.6.2   Diagnosed heating rates**

A complementary set of heating rates is diagnosed using a subset of the zonal-mean dynamical quantities introduced above. These heating rates have been calculated based on analysis of the zonal-mean thermodynamic equation:

$$\frac{\partial \overline{\theta}}{\partial t} + \frac{\overline{v}}{a} \frac{\partial \overline{\theta}}{\partial \phi} + \overline{\omega} \frac{\partial \overline{\theta}}{\partial p} + \frac{1}{a \cos \phi} \frac{\partial (\overline{v' \theta'} \cos \phi)}{\partial \phi} + \frac{\partial (\overline{\omega' \theta'})}{\partial p} + \overline{\mathcal{X}} = \overline{\frac{\theta}{c_p T} \dot{Q}}, \tag{12}$$





and are provided together with dynamical heat transport terms. Terms on the left-hand side that are functions of potential temperature are obtained from the corresponding terms expressed as functions of temperature (see Tables 3 and 4). For example, vertical fluxes of potential temperatures are obtained from $\overline{\omega'T'}$ using the identity $\overline{\omega'\theta'} = \overline{\omega'T'}\,(p/p_0)^{-\kappa}$, with $\kappa$ approximated as 0.286 as in Eq. 1. The term $\overline{\mathcal{X}}$, which can be computed as a residual by substituting the model-generated diabatic heating into Eq. 12, represents the cumulative effects of assimilation increments and numerical errors.

**Table 8.** Diabatic and dynamical heating diagnostics.

| Variable | Expression | Units |
|---|---|---|
| Time rate of change in potential temperature | $\frac{\partial \overline{\theta}}{\partial t}$ | $\mathrm{K\,day}^{-1}$ |
| Meridional advection of potential temperature | $\frac{\overline{v}}{a}\frac{\partial \overline{\theta}}{\partial \phi}$ | $\mathrm{K\,day}^{-1}$ |
| Vertical advection of potential temperature | $\overline{\omega}\frac{\partial \overline{\theta}}{\partial p}$ | $\mathrm{K\,day}^{-1}$ |
| Meridional eddy term | $\frac{1}{a\cos\phi}\frac{\partial(\overline{v'\theta'}\cos\phi)}{\partial \phi}$ | $\mathrm{K\,day}^{-1}$ |
| Vertical eddy term | $\frac{\partial(\overline{\omega'\theta'})}{\partial p}$ | $\mathrm{K\,day}^{-1}$ |
| Estimated total diabatic heating | $\frac{D\overline{\theta}}{Dt}$ | $\mathrm{K\,day}^{-1}$ |
| Residual | $\overline{\mathcal{X}}$ | $\mathrm{K\,day}^{-1}$ |

Table 8 lists the diabatic and dynamical heating terms calculated based on Eq. 12. As with the model-generated diabatic heating rates, all terms are provided as functions of potential temperature in units of $\mathrm{K\,day}^{-1}$. Although these diagnostics are based on the core variables and covariance terms, they are constructed to be valid at 03, 09, 15, and 21Z to match the model-generated diabatic heating rates. The time derivative $\partial \overline{\theta}/\partial t$ is calculated as a central difference, while the meridional and pressure derivatives are calculated by applying the numerical methods described in Section 3.1 to quantities averaged over the two time steps bracketing each window.

## 4 Comparison of the OG and CG data sets

All diagnostics are provided on two distinct grids. For the original grid (OG) data set, the diagnostics described in Section 3 are calculated on the grid specific to the corresponding reanalysis (see Table 1). Differences in the diagnostics amongst reanalyses are therefore influenced by numerical resolution. Small-scale processes could affect the computation of co-variance terms, with reanalysis products on a finer grid potentially providing more accurate representations. These data, in addition, cannot be compared directly across reanalyses due to the variety of grids unless the user performs area integrals or interpolations. By contrast, the common grid (CG) data set provides all diagnostics on a common $2.5°\times2.5°$ grid and uses only those pressure levels common to all fourteen reanalyses (Table 2). Reanalyses that provide data at finer resolutions are regridded in the



horizontal dimension using bilinear interpolation. All diagnostics are calculated after this interpolation is performed. This approach enables direct comparison of all reanalyses, but the interpolation may introduce additional numerical errors.

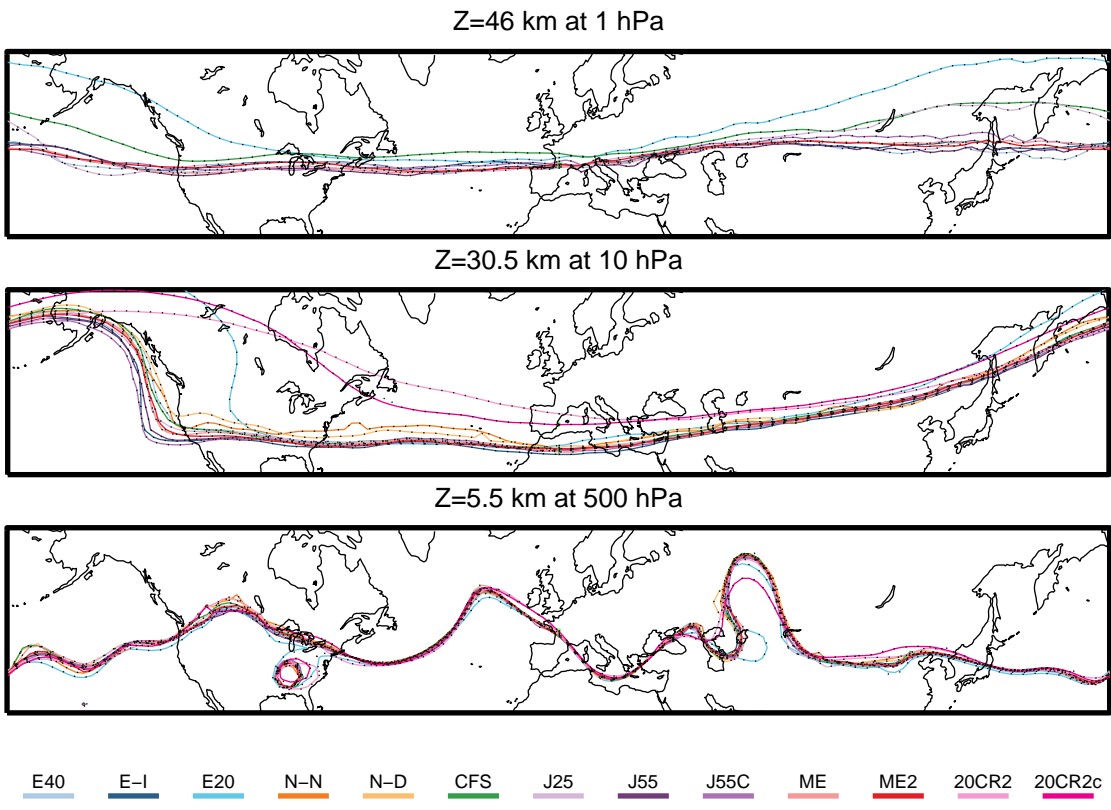

**Figure 2.** Selected geopotential height contours on January 1 1980 at 00 UTC for three isobaric surfaces: 1 hPa (top; $Z = 46$ km), 10 hPa (center; $Z = 30.5$ km), and 500 hPa (bottom; $Z = 5.5$ km). Contours are displayed for each reanalysis according to color legend. Contours based on the OG data set are shown using solid lines. Black dots are added to contours based on the CG data set. The two sets of contours are indistinguishable for this case. Due to space constraints, these longitude-dependent fields are not included in the core data set.

The impact of the grid transformation on variables provided in these data sets is tested for some selected diagnostics. Figure 2 shows geopotential height contours for all reanalyses using data from the OG and CG data sets at 00UTC 1 January

5    1980. Contours based on OG and CG data for each reanalysis are virtually indistinguishable from each other. Inter-reanalysis differences are far larger than discrepancies between grid types. Figure 3 shows the vertical profile of zonal wind averaged over the Northern Hemisphere high latitudes at 00UTC 1 January 1980. Despite some small differences, results based on the OG and CG data sets are generally similar. Discrepancies between the two grid types can be partly attributed to uncertainties in the interpolation procedure; however, these discrepancies are again smaller than differences across reanalyses.

10    The differences between the CG and OG profiles shown in Fig. 3 can be attributed to latitudinal resolution. The number of grid points included in the latitude band used to compute the average differs between CG and OG data based on the same



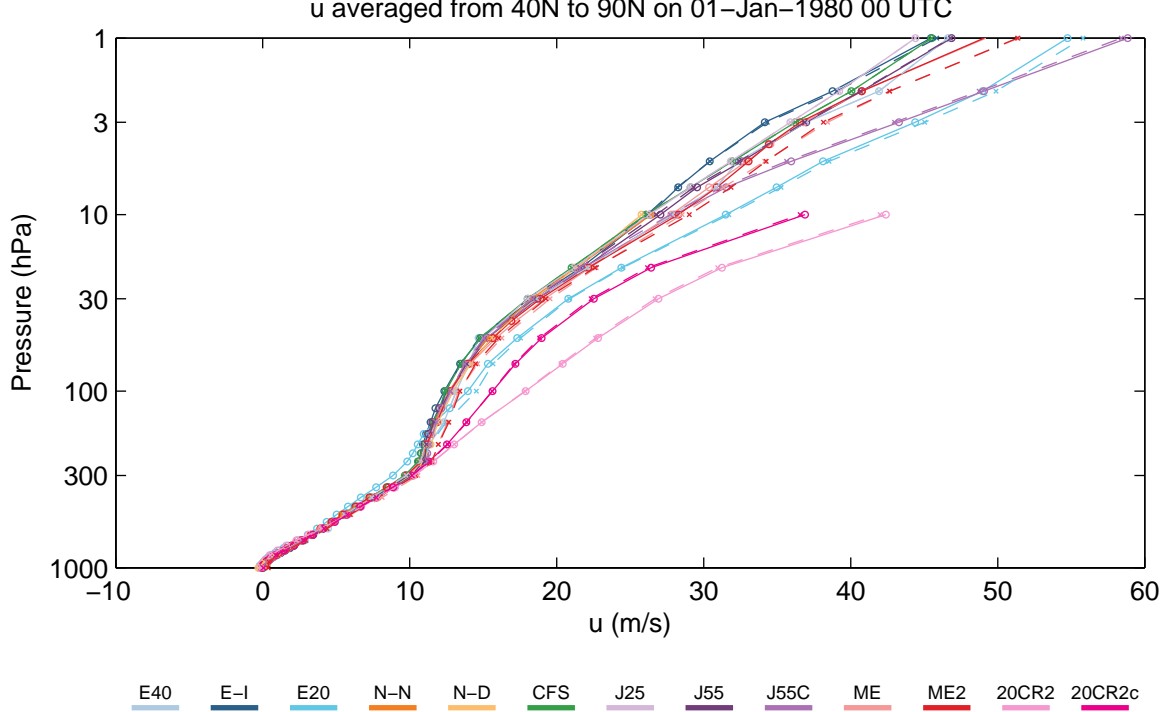

**Figure 3.** Vertical profiles of zonal wind averaged from 40°N to 90°N at 00UTC 1 January 1980. Profiles are shown for both the common grid (solid, x) and original grid (dashed, o) data based on different reanalyses (colors). Tick marks (x, o) indicate pressure levels included in the corresponding grid.

reanalysis data set. These differences can influence budget averages. Figure 4 shows the detailed zonal-mean distribution of zonal wind for both grid types and all reanalyses. The OG and CG data sets are virtually indistinguishable from this perspective, indicating that the interpolation used to create the CG data set has little influence on zonal-mean quantities.

Although zonal-mean quantities are largely insensitive to grid spacing and interpolation, flux terms may be more sensitive.
5 Figure 5 shows the vertical profile of EP flux divergence, a quantity that depends on both the horizontal resolution (for computing heat and momentum fluxes) and the vertical resolution (for computing vertical derivatives). Again, differences between EP flux divergence computed using CG data and EP flux divergence computed using OG data are typically small relative to differences across reanalyses. As above, some of the differences between the CG and OG diagnostics are due to the different numbers of points that go into the meridional average, but differences in vertical resolution also play an important role. The
10 latter is particularly apparent in the differences between the CG and OG profiles for MERRA and MERRA-2 between 30 and 100 hPa. The OG and CG profiles for NCEP–NCAR and NCEP–DOE, for which the OG and CG grids are identical in this latitude range, are virtually the same at all levels.

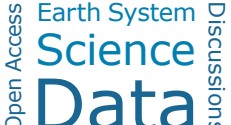

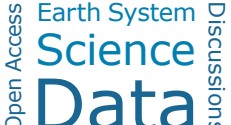

**Figure 4.** Zonal wind as a function of latitude for the 1 hPa (top), 10 hPa (second from top), 100 hPa (second from bottom), and 300 hPa (bottom) pressure levels at 00UTC 1 January 1980. Values are shown for both the common grid (solid, x) and original grid (dashed, o) data based on different reanalyses (colors). Tick marks (x, o) indicate latitude points included in the corresponding grid.




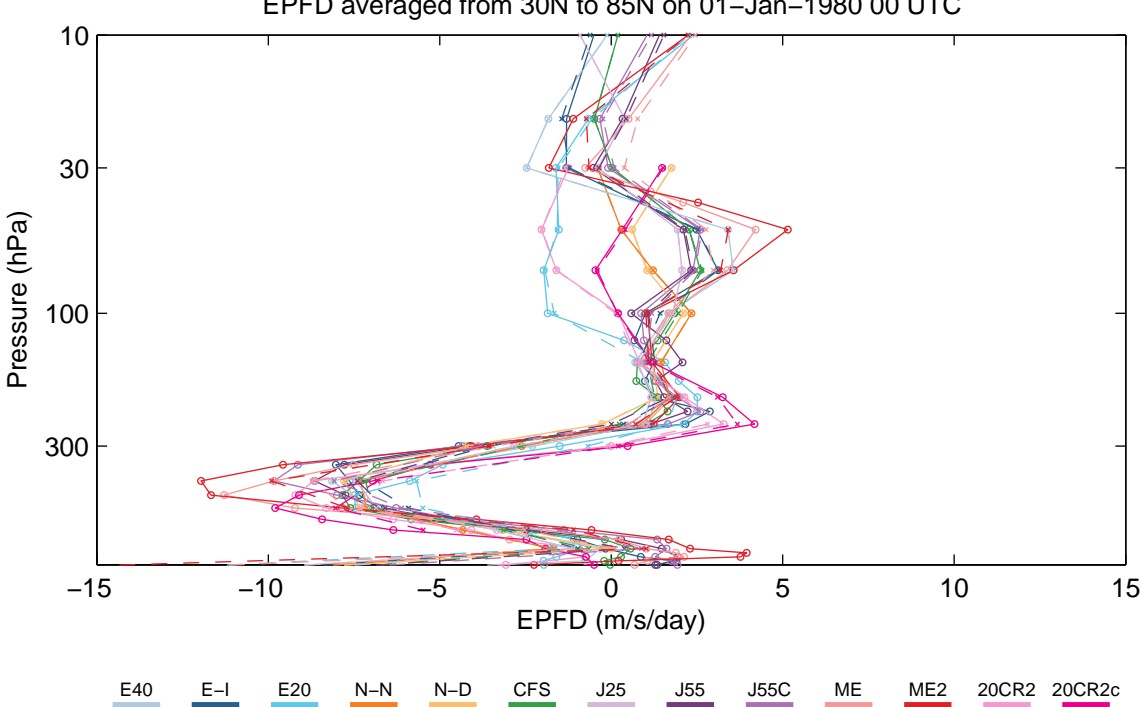

**Figure 5.** Vertical profiles of EP flux divergence averaged over 30°N to 85°N at 00UTC 1 January 1980. Profiles are shown for both the common grid (solid, x) and original grid (dashed, o) data based on different reanalyses (colors). Tick marks (x, o) indicate pressure levels included in the corresponding grid. Note that values for 20CR and ERA-20C (both surface-input reanalyses; see Fujiwara et al., 2017) are negative around 30–10 hPa.

Figure 6 shows EP flux divergence as a function of latitude. Although the values of this diagnostic are typically similar between the CG and OG data sets, they differ substantially in some locations. This is especially evident for ERA-20C at 300 hPa. Such differences likely result from differences in the relative contributions of small-scale eddies, which are enhanced when computations are performed using OG data and reduced when computations are performed using CG data (note also

5 that ERA-20C has the finest OG grid spacing amongst the reanalyses; Table 1). EP flux divergence also varies substantially amongst reanalyses near the pole. These inter-reanalysis differences are likely related to differences in representations of eddy fluxes at the poles amongst reanalyses. We therefore recommend that users of this data set be cautious in interpreting behavior near the boundaries and avoid using certain diagnostics in these regions.

The sensitivity of momentum diagnostics to numerical resolution has been evaluated separately in both horizontal and

10 vertical dimensions by Martineau et al. (2016). Although enhanced vertical resolution improves dynamical consistency (i.e. the ability to explain the wind tendency as a function of the forcing terms) in the upper troposphere and lower stratosphere, gains in dynamical consistency in the middle stratosphere are mainly achieved by reducing the horizontal grid spacing. However,



**Figure 6.** EP flux divergence as a function of latitude for the 1 hPa (top), 10 hPa (second from top), 100 hPa (second from bottom), and 300 hPa (bottom) pressure levels at 00UTC 1 January 1980. Values are shown for both the common grid (solid, x) and original grid (dashed, o) data based on different reanalyses (colors). Tick marks (x, o) indicate latitude points included in the corresponding grid.





these improvements were merely incremental in both cases. The small differences in grid spacing between the OG and CG data sets are thus not expected to substantially affect the conclusions of studies that use these data sets.

Overall, differences between the OG and CG diabatic heating diagnostics are similar to those for other variables: very small in zonal-mean fields, slightly larger for area averages, and typically much smaller than inter-reanalysis differences. The latter two features are illustrated in Fig. 7a, which shows time series of area-mean model-generated diabatic heating at 50 hPa averaged over 60°N to 90°N from January through March 2009 (Manney et al., 2009; Harada et al., 2010). These time series include the sharp intensification and subsequent decay in diabatic cooling associated with a stratospheric sudden warming that occurred around 24 January 2009. Differences between the OG and CG data sets are negligible through most of the period shown, except for the weeks immediately following the sudden warming. Although qualitative variations are consistent between the two data sets, CG cooling rates are enhanced relative to OG cooling rates during portions of this period by magnitudes approaching $0.2\,\mathrm{K\,day^{-1}}$. Users should be aware of the potential for these types of quantitative biases when using the OG and/or CG data sets to study temporal variations in area-mean quantities.

Figure 7b shows a similar time series of heating rates diagnosed from the core dynamical variables using the zonal-mean thermodynamic equation. Although the qualitative behavior of the diagnosed heating rates is similar to that of the model-generated forecasts through most of the comparison period, the two estimates differ substantially in the lead-up to the sudden warming and the days immediately afterward. Three aspects of these diagnosed heating rates are worth noting here. First, the diagnosed heating rates are considerably noisier than the model-generated heating rates, particularly at 6-hourly time resolution. This noise arises largely from variance in the vertical velocity $\omega$ (accruing at least in part from using the average of two instantaneous values bracketing the 6-h time step rather than average values across the time step), as well as numerical errors during the diagnostic step. For practical applications, the noise can be reduced by applying a rolling mean. The rolling mean is applied using a Hamming window spanning two days (nine time steps) for the time series shown in Fig. 7b. Second, differences between the OG and CG data sets are much larger for the diagnosed heating rates than for the model-generated heating rates. For the time series shown in Fig. 7b, these differences are especially pronounced during the period leading up to the sudden warming (as large as $0.5{\sim}0.7\,\mathrm{K\,day^{-1}}$ in the diagnosed total heating rate). Several of the terms on the left-hand side of Eq. 12 increase sharply in absolute magnitude during this period, with the differences between the OG and CG representations of these terms increasing at the same time. Third, the diagnosed heating rates do not extend over the entire polar cap. Edge effects eliminate the data at 90° and adversely impact the quality of the data at 87.5°; note, however, that calculating average model-generated heating rates over the 60°N–85°N domain has little influence on the time series shown in Fig. 7a.

## 5 Data usage and availability

The S-RIP zonal-mean data set of reanalyses on pressure levels provides pre-processed zonal-mean diagnostics using unified NetCDF-4 classic file format. The main purpose of making this data set publicly available is to reduce the workload of researchers contributing to the S-RIP project by providing diagnostics that are commonly needed for reanalysis intercomparison, particularly in the middle atmosphere. The provision of pre-processed data will also save users the need to download and



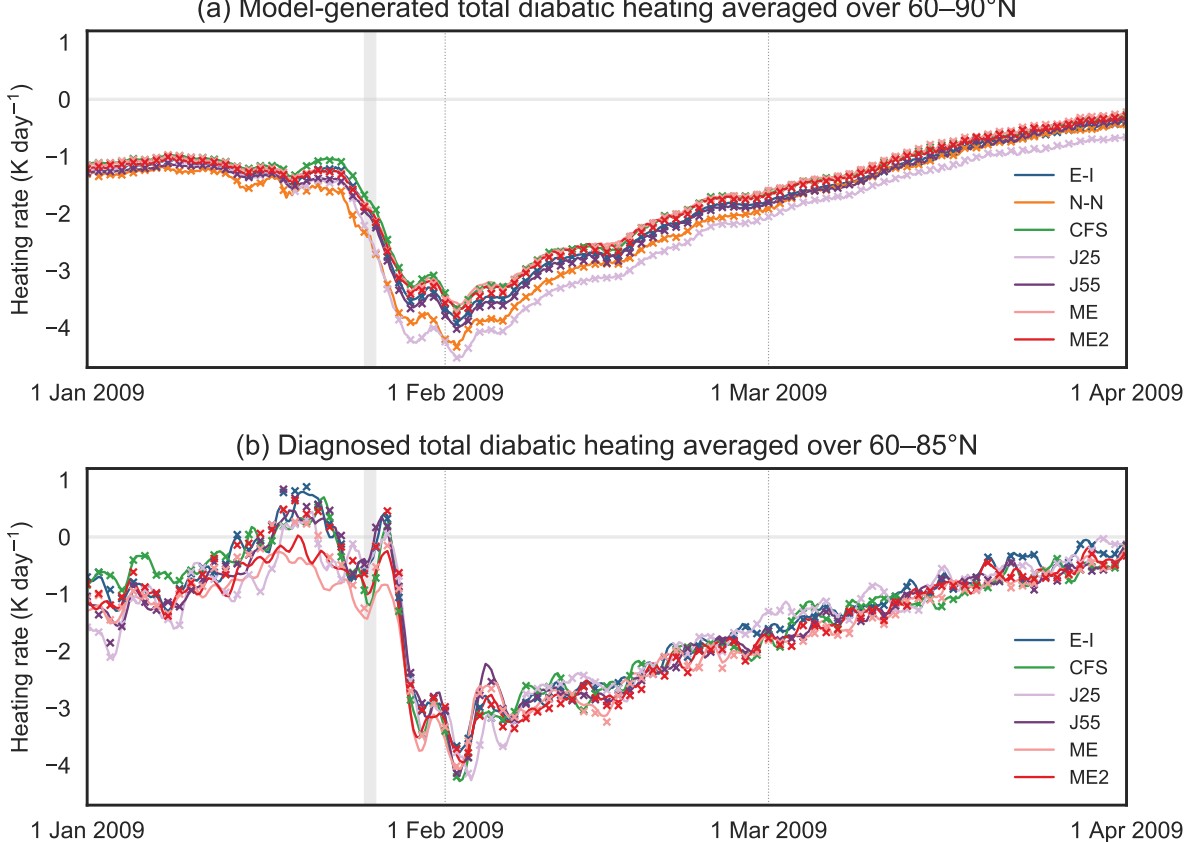

**Figure 7.** Time series of total diabatic heating on the 50 hPa isobaric surface based on (a) model-generated forecasts averaged over 60°N to 90°N and (b) analysis of the zonal-mean thermodynamic equation averaged over 60°N to 85°N. Time series are shown for the period between 00UTC 1 January 2009 and 00UTC 1 April 2009. Note that a 2-day rolling mean is applied to the diagnosed heating rates. Values are shown at 6-h intervals for the original grid (lines) and 24-h intervals for the common grid (x; 03UTC every day) based on reanalyses with available data for the selected time period and isobaric level (colors). 24 January is marked by the vertical shaded region.





store dozens of terabytes of data. Producing the data set locally using a standardized set of computer codes ensures that the diagnostics are consistent amongst the reanalyses.

The dynamical (Martineau, 2017) and diabatic (Wright, 2017) components of the data set are archived and maintained by the Centre for Environmental Data Analysis (CEDA) and have been made CF-compliant when possible. All NetCDF files are

fully annotated with descriptions of variables and units. A user manual describing the files in detail is provided. The data set is bound to evolve in the future as new reanalysis products are introduced, and is also being updated to include additional data as reanalyses are extended in time. Since this data set is intended to serve the needs of the S-RIP community, it may be extended to include additional diagnostics as dictated by user requirements.

## 6  Summary

The S-RIP zonal-mean data set of reanalyses on pressure levels aims to facilitate the comparison of reanalysis data sets for the S-RIP community and the general atmospheric science community at large. In its current iteration, the data set includes 14 reanalyses and ancillary products from multiple research institutes. It covers the satellite era (1979–present) and extends backward in time to 1958 when data is available. Diagnostics provided include zonal-mean variables, diabatic heating, covariance and variance terms, and complete diagnostics from the Eulerian-mean and transformed Eulerian mean momentum equations.

The diagnostics are provided on two grids, the original grid (OG) where diagnostics are performed on the original files acquired from each reanalysis center, and the common grid (CG) where data is interpolated to a unified grid before advanced diagnostics are performed. The data set will grow in time to include more reanalyses and variables, as dictated by the evolving needs of the S-RIP community.

**Appendix A: Data access**



**Table A1.** Websites and dates of access for core reanalysis variables.

| Name | URL or DOI | Date accessed |
|------|-----------|---------------|
| ERA-40 | http://www.ecmwf.int/en/forecasts/datasets/era-40-dataset-sep-1957-aug-2002 | 2010-11-24 |
| ERA-Interim | http://www.ecmwf.int/en/research/climate-reanalysis/era-interim | 2017-09-21 |
| ERA-20C | https://doi.org/10.5065/D6VQ30QG | 2015-12-31 |
| NCEP-NCAR | http://www.esrl.noaa.gov/psd | 2017-09-21 |
| NCEP-DOE | http://www.esrl.noaa.gov/psd | 2017-10-03 |
| CFSR | http://dx.doi.org/10.5065/D69K487J | 2017-10-04 |
| 20CR (v2) | https://www.esrl.noaa.gov/psd/data/gridded/data.20thC_ReanV2.html | 2013-07-02 |
| 20CR (v2c) | http://www.esrl.noaa.gov/psd/data/gridded/data.20thC_ReanV2c.html | 2016-04-05 |
| JRA-25 | http://rda.ucar.edu/datasets/ds625.0/ | 2017-10-05 |
| JRA-55 | http://dx.doi.org/10.5065/D6HH6H41 | 2017-10-26 |
| JRA-55C | https://doi.org/10.5065/D67H1GNZ | 2017-11-05 |
| JRA-55AMIP | https://doi.org/10.5065/D6TB14ZD | 2015-12-09 |
| MERRA | http://disc.sci.gsfc.nasa.gov/mdisc/ | 2017-10-04 |
| MERRA-2 | http://doi.org/10.5067/QBZ6MG944HW0 | 2017-07-05 |

**Table A2.** Websites and dates of access for model-generated reanalysis diabatic heating products.

| Name | URL or DOI | Date accessed |
|------|-----------|---------------|
| ERA-40 | http://apps.ecmwf.int/datasets/data/era40-daily[a] | 2017-08-02 |
| ERA-Interim | http://apps.ecmwf.int/datasets/data/interim-full-daily[a] | 2017-08-07 |
| NCEP-NCAR | http://rda.ucar.edu/datasets/ds090.0 | 2017-08-25 |
| CFSR | https://doi.org/10.5065/D69K487J | 2017-07-20 |
| JRA-25 | http://rda.ucar.edu/datasets/ds625.0 | 2017-07-25 |
| JRA-55 | https://doi.org/10.5065/D6HH6H41 | 2017-08-04 |
| MERRA | https://doi.org/10.5067/RP02UMM6LH1B | 2017-08-01 |
| | https://doi.org/10.5067/DNZTCFMAG3FW | |
| MERRA-2 | https://doi.org/10.5067/9NCR9DDDOPFI | 2017-08-27 |
| | https://doi.org/10.5067/3UGE8WQXZAOK | |

[a] Data accessed via the ECMWF Web API (https://software.ecmwf.int)



## Appendix B: List of abbreviations

| | |
|---|---|
| 20CR2 | 20th Century Reanalysis of NOAA and CIRES Version 2 |
| 20CR2c | 20th Century Reanalysis of NOAA and CIRES Version 2c |
| AMIP | Atmospheric Model Intercomparison Project |
| CFS | (CFSR) Climate Forecast System Reanalysis of NCEP |
| CG | Common Grid |
| CIRES | Cooperative Institute for Research in Environmental Science at the University of Colorado, Boulder |
| DOE | Department of Energy |
| E-I | (ERA-Interim) ECMWF interim reanalysis |
| E20 | (ERA-20C) ECMWF 20th century reanalysis |
| E40 | (ERA-40) ECMWF 40-year reanalysis |
| J25 | (JRA-25) Japanese 25-year Reanalysis |
| J55 | (JRA-55) Japanese 55-year Reanalysis |
| J55C | (JRA-55C) Japanese 55-year Reanalysis assimilating Conventional observations only |
| N-D | (NCEP-DOE) NCEP/DOE reanalysis |
| N-N | (NCEP-NCAR) NCEP/NCAR reanalysis |
| ME | (MERRA) Modern Era Retrospective-Analysis for Research |
| ME2 | (MERRA-2) Modern Era Retrospective-Analysis for Research version 2 |
| NASA | National Aeronautics and Space Administration |
| NCAR | National Center for Atmospheric Research |
| NCEP | National Centers for Environmental Prediction of the NOAA |
| NOAA | National Oceanic and Atmospheric Administration |
| OG | Original Grid |
| S-RIP | SPARC-Reanalysis Intercomparison Project |
| SPARC | Stratosphere-troposphere Processes And their Role in Climate |
| TEM | Transformed Eulerian Mean |

*Acknowledgements.* We appreciate the support of the British Atmospheric Data Centre (BADC) of the UK Centre of Environmental Data Analysis (CEDA) for hosting the data set on their servers. We thank the reanalysis centres for providing support and access to data products. We also thank Amy Butler, Andrew Orr, Peter Hitchcock and Edwin Gerber for bringing coding errors and missing data to our attention, James
5   Anstey and the BADC for helping to secure storage space for archiving and processing the raw data, and Chi-Fan Shih for assistance with resolving data access problems. Masatomo Fujiwara's contribution was financially supported in part by the Japan Society for the Promotion of Science (JSPS) through Grants-in-Aid for Scientific Research (26287117 and 16K05548). Patrick Martineau was partially supported by the US CAREER grant AGS-1742178 that was awarded to Gang Chen through Cornell University and UCLA. Jonathon Wright and Nuanliang

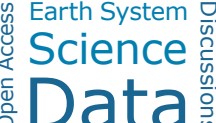

Zhu were supported by the Ministry of Science and Technology of the People's Republic of China (2017YFA0603900) and the National Natural Science Foundation of China (41761134097).



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
