# Peer review of "Zonal-mean data set of global atmospheric reanalyses on pressure levels"

_Earth System Science Data, 2018_

## Referee Comment (RC1) · Anonymous Referee #1 · 24 Jul 2018

Review of the paper:

"Zonal-mean data set of global atmospheric..."

written by Patrick Martineau et al.,

**General:**
The paper presents a very comprehensive analysis comparing different re-analysis products with respect to their representation of the zonally-averaged basic dynamical quantities as well as to their representation of more sophisticated parameters like wave-forcing (EP flux and its divergence) or the diabatic 2d residual circulation. This

comparison was performed within the SPARC-Reanalysis Intercomparison Project (S-RIP) and the provided results are archived and available for the scientific community. The presented analysis is very clean and covers the issue from all different angles. Thus, I would like to recommend this paper for publishing in Earth System Science Data (ESSD) with only some minor points listed below.

**Minor comments:**

- Captions of figures 3-6
  It looks for me that solid lines are denoted with "o" and dashed lines with "x", i.e. vice verse than the explanation in the manuscript

- P15 L2
  "this is especially evident for ERA-20C at 300 hPa" - little bit difficult to see it

- P15 L6
  "EP flux divergence also varies substantially amongst reanalyses" - maybe you should add "(not shown)"

- P18 Figure 7
  It would be also nice to plot the difference between Fig 7b and Fig 7a measuring the strength of the influence of the assimilation procedure. I.e. before during and after the sudden warming, the strongest assimilation increments are necessary to "keep the model on track".

- P 17, Data usage
  It would be also great to get zonally-resolved monthly means of some of quantities discussed here like temperature, wind, GPH, but also the zonally resolved EP flux divergence as described in Plumb, JAS, 1985.

---

## Referee Comment (RC2) · Anonymous Referee #2 · 27 Aug 2018

This work presents a new dataset that provides zonal average diagnostics from a number of reanalysis. The data are interpolated to a common set of pressure levels at 2.5° resolution in latitude. The choice of pressure levels and the name of variables follow the CF convention, indicating that the dataset is intended to be used by the large community of climate change studies. The data are associated to a couple of DOIs and are available on CEDA. I checked that they can be actually accessed and can be read with standard netcdf tools. As the work is done under the framework of the S-RIP project, the dataset provides a few more levels than the CF standard set in the stratosphere. The paper results from the juxtaposition of two independent works on dynamical variables and on the diabatic heating rates which are processed separately. The paper does not address the matter of comparing the reanalyses and only study the effect of

interpolating from the so-called original grid (see below) to the common grid. This is OK since such a comparison on all the variables would by itself deserve several studies and it is important to advertise quite rapidly this dataset so that it can be used in the CFMIP-6 intercomparison. I am therefore favourable to publish the manuscript with a limited number of reservations as follows:

- The main reservation is about the notion of original grid in the paper which is indeed the pressure grid on which the reanalyses centres are projecting their data for dissemination. The true original grid of the model - on which data are also available, at least for modern reanalysis - is not that one but usually a hybrid grid in the vertical and various meshes or spherical harmonics truncations in the horizontal. Therefore what the authors call the original grid is not the original grid but results from interpolation on a reduced set of levels and grid points. It is plausible that the effect is small on most diagnostics presented here but this double interpolation leading to the common grid should be mentioned quite clearly. The data will be probably used by some people who only have a vague notion of how an atmospheric model is built and it is important to tell them what they get.

- The noise due to this first interpolation is hard to estimate from the data presented here but is not necessarily smaller than that between the OG and the CG.

- I do not see the need to show two figures on the zonal wind comparison. Figure 4 could be omitted without any loss.

- Even if reanalyses intercomparison is not the topic of the paper, it could be mentioned that the reanalyses for which altitude data depart from the main group are those which rely only on surface observations.

- It is not surprising that products of zonal anomalies exhibit more noise due to interpolation than the main variables. My opinion is that such quantities should be calculated at the highest available resolution (on a pressure grid with the same resolution as the true original grid) and then interpolated to the target grid.
- It is a pity that clear sky heating rates are not included. They are not provided by all reanalysis and they actually differ much less than the all sky heating rates but they could be useful to calculate the cloud radiative heating rates.

- It is also a pity that no cloud diagnostics like cloud cover, liquid and ice water content, are provided. This is where reanalyses and climate models are exhibiting the largest discrepancies among them and what can explain the discrepancies in the heating rates.

- It is mentioned that new reanalyses will be included when available. It ERA5, which is already available, will be soon included, it is worth mentioning it.

- Perhaps the name of the variables as they appear in the file could be added in tables 3 to 6 as this is not always obvious. There is an additional description file associated with the dataset but it could be useful to have all the information in one single location.

I have also a few comments related to the way data are presented on CEDA:

- It would be useful that the two components of the dataset are cross-linked on the two DOI landing pages. That is not the case presently.

- The heating rates are described as being temperature tendencies and in K s-1 in the diabatic landing page while they are actually potential temperature tendencies and in K day-1. Please fix this confusion.

- The diagnostic quantities described in section 3.6.2 are not available from CEDA. Make them available or remove them from the paper.

- It would be convenient that the dynamical and diabatic dataset have exactly the same format. The diabatic netcdf files contain an extra longitude dimension of size 1.

---

## Author Comment (AC1) · 25 Sep 2018

We are thankful to the reviewer for these comments which help us improve the quality of our manuscript. Our response to each comment is written below.

**Anonymous Referee #1**

General:

The paper presents a very comprehensive analysis comparing different re-analysis products with respect to their representation of the zonally-averaged basic dynamical quantities as well as to their representation of more sophisticated parameters like wave-forcing (EP flux and its divergence) or the diabatic 2d residual circulation. This comparison was performed within the SPARC-Reanalysis Intercomparison Project (SRIP) and the provided results are archived and available for the scientific community. The presented analysis is very clean and covers the issue from all different angles. Thus, I would like to recommend this paper for publishing in Earth System Science Data (ESSD) with only some minor points listed below.

Minor comments:

1. Captions of figures 3-6 It looks for me that solid lines are denoted with "o" and dashed lines with "x", i.e. vice verse than the explanation in the manuscript

   Thank you very much for finding this mistake. The figure captions are corrected.

2. P15 L2 "this is especially evident for ERA-20C at 300 hPa" - little bit difficult to see it

   To improve clarity, we now indicate that the difference we are trying to highlight here is located at around 37°N.

3. P15 L6 "EP flux divergence also varies substantially amongst reanalyses" - maybe you should add "(not shown)"

   This is shown in Figure 5 (previously Figure 6) for the North Pole. EP-flux divergence values can be quite different among data sets, especially at 300 hPa and 10 hPa.

4. P18 Figure 7 It would be also nice to plot the difference between Fig 7b and Fig 7a measuring the strength of the influence of the assimilation procedure. I.e. before during and after the sudden warming, the strongest assimilation increments are necessary to "keep the model on track".

   We have added a third panel to this figure (now Figure 6) showing the magnitude of the residual term. Although the residual term represents one estimate of the assimilation increment, its interpretation as such is not as straightforward as it initially seems. We have added a paragraph to clarify this and the reasons behind it, along with a figure showing the breakdown of the MERRA-2 heat budget into time rate of change, dynamics, residual/analysis, and diabatic terms (Figure 7). We select MERRA-2 for this illustration because it provides outputs for the dynamical, analysis, and diabatic terms, allowing us to calculate the time rate of change independently as a residual and thus permitting a direct comparison with the terms in our diagnosed heat budget.

5.  P 17, Data usage It would be also great to get zonally-resolved monthly means of some of quantities discussed here like temperature, wind, GPH, but also the zonally resolved EP flux divergence as described in Plumb, JAS, 1985.

    A zonally-resolved monthly data set was prepared and is available to the contributors of the S-RIP project but has not been made publicly available at this time. One has also to consider the added-value of providing monthly averages of basic fields on pressure levels considering that many reanalysis centers already provide monthly mean data. Advanced diagnostics such as the Plumb flux are certainly interesting to provide and we will consider doing so in a future iteration of the data set or in a separate data set.

---

## Author Comment (AC2) · 25 Sep 2018

We are thankful to the reviewer for these comments which help us improve the quality of our manuscript. Our response to each comment is written below.

**Anonymous Referee #2**

This work presents a new dataset that provides zonal average diagnostics from a number of reanalysis. The data are interpolated to a common set of pressure levels at 2.5 resolution in latitude. The choice of pressure levels and the name of variables follow the CF convention, indicating that the dataset is intended to be used by the large community of climate change studies. The data are associated to a couple of DOIs and are available on CEDA. I checked that they can be actually accessed and can be read with standard netcdf tools. As the work is done under the framework of the S-RIP project, the dataset provides a few more levels than the CF standard set in the stratosphere. The paper results from the juxtaposition of two independent works on dynamical variables and on the diabatic heating rates which are processed separately. The paper does not address the matter of comparing the reanalyses and only study the effect of interpolating from the so-called original grid (see below) to the common grid. This is OK since such a comparison on all the variables would by itself deserve several studies and it is important to advertise quite rapidly this dataset so that it can be used in the CFMIP-6 intercomparison. I am therefore favourable to publish the manuscript with a limited number of reservations as follows:

1. The main reservation is about the notion of original grid in the paper which is indeed the pressure grid on which the reanalyses centres are projecting their data for dissemination. The true original grid of the model - on which data are also available, at least for modern reanalysis - is not that one but usually a hybrid grid in the vertical and various meshes or spherical harmonics truncations in the horizontal. Therefore what the authors call the original grid is not the original grid but results from interpolation on a reduced set of levels and grid points. It is plausible that the effect is small on most diagnostics presented here but this double interpolation leading to the common grid should be mentioned quite clearly. The data will be probably used by some people who only have a vague notion of how an atmospheric model is built and it is important to tell them what they get. - The noise due to this first interpolation is hard to estimate from the data presented here but is not necessarily smaller than that between the OG and the CG.

   We agree with the reviewer that we should clarify that "original grid" here does not refer to the grid on which the model was run, and that the Original Grid data have already been interpolated to coarser grids for data distribution. This information was previously included in the introduction but is now emphasized in sections 3 and 4 as well.

2. I do not see the need to show two figures on the zonal wind comparison. Figure 4 could be omitted without any loss.

   We have removed Fig. 4 from the manuscript.

3. Even if reanalyses intercomparison is not the topic of the paper, it could be mentioned that the reanalyses for which altitude data depart from the main group are those which rely only on surface observations.

   We agree that it is useful to indicate that outliers are datasets that do not assimilate conventional observations. We now do so when such differences are first shown in our discussion of Fig. 2.

4. It is not surprising that products of zonal anomalies exhibit more noise due to interpolation than the main variables. My opinion is that such quantities should be calculated at the highest available resolution (on a pressure grid with the same resolution as the true original grid) and then interpolated to the target grid.

   Our intention with the Common Grid dataset is to allow a fair comparison of reanalysis data sets that is not affected by grid resolution. This is why we performed bilinear interpolation to a common grid before computing eddy terms. We explain and justify this approach in the introduction and in section 4.

   We do agree with the reviewer that to best evaluate eddy processes, one has to compute eddy terms with the finest resolution before interpolating to a coarser grid. Although we do not provide such data set, it can be computed by users starting from the Original Grid data set. Simple one-dimensional interpolation in latitude can be used instead of the more computationally demanding bilinear interpolation we have used for the common grid data set. If one wishes to compare variables integrated over an area, only integration is needed. We agree with the reviewer that some users may want to compare reanalysis datasets including the effect of resolution. We therefore mention how to proceed in section 4.

5. It is a pity that clear sky heating rates are not included. They are not provided by all reanalysis and they actually differ much less than the all sky heating rates but they could be useful to calculate the cloud radiative heating rates.

   Unfortunately, only the ECMWF and GMAO reanalyses have archived the clear-sky radiation tendency terms. We prefer to prioritize consistency over completeness in this initial version of the diabatic data set, and so have opted not to include variables that are provided for only half of the considered reanalyses.

6. It is also a pity that no cloud diagnostics like cloud cover, liquid and ice water content, are provided. This is where reanalyses and climate models are exhibiting the largest discrepancies among them and what can explain the discrepancies in the heating rates.

   We agree that differences in cloud fields and parameterizations are behind many of the largest discrepancies among the reanalyses. However, as with the clear-sky radiative heating rates, we need to solve issues of consistency amongst the reanalyses to include these terms. For example, some of the reanalyses do not provide vertical profiles of cloud cover, and several do not distinguish between liquid and ice water content. Although we do not include them now, we will consider options for adding cloud and composition variables in a future iteration of the diabatic data set, perhaps along with additional derived variables such as clear-sky radiative heating.

7. It is mentioned that new reanalyses will be included when available. It ERA5, which is already available, will be soon included, it is worth mentioning it.

Thank you for this comment. The dataset has been produced for ERA5 for some selected diagnostics from 2008 to 2016, but we are waiting for the full release to make it public. We now mention this future extension in section 5.

8.  Perhaps the name of the variables as they appear in the file could be added in tables 3 to 6 as this is not always obvious. There is an additional description file associated with the dataset but it could be useful to have all the information in one single location.

    We have added variable names to the tables.

I have also a few comments related to the way data are presented on CEDA:

9.  It would be useful that the two components of the dataset are cross-linked on the two DOI landing pages. That is not the case presently.

    The data-sets are now cross-linked together in the "Related Records" tab.

10. The heating rates are described as being temperature tendencies and in K s-1 in the diabatic landing page while they are actually potential temperature tendencies and in K day-1. Please fix this confusion.

    Attributes of variables have been updated on the CEDA page. All tendencies are now listed as potential temperature tendencies.

11. The diagnostic quantities described in section 3.6.2 are not available from CEDA. Make them available or remove them from the paper.

    These quantities are provided in the files but are not listed on the CEDA description page due to a bug affecting the archiving process. CEDA is actively working to resolve the issue.

12. It would be convenient that the dynamical and diabatic dataset have exactly the same format. The diabatic netcdf files contain an extra longitude dimension of size 1.

    We will resolve this inconsistency between the two data sets when preparing the next release.